# Impact of $^{18}$F-FDG PET/CT in Predicting Recurrence in Neurolymphomatosis

Megumi Uemichi [1], Kota Yokoyama [1,*], Junichi Tsuchiya [1], Toshiki Terao [2], Youichi Machida [3], Kosei Matsue [2] and Ukihide Tateishi [1]

1    Department of Diagnostic Radiology, Tokyo Medical and Dental University, Tokyo 113-8510, Japan; jsdrmbda_16@yahoo.co.jp (M.U.); tuwu11@gmail.com (J.T.); ttisdrnm@tmd.ac.jp (U.T.)
2    Kameda Medical Center, Division of Hematology/Oncology, Department of Internal Medicine, Chiba 296-0041, Japan; terao.toshiki@kameda.jp (T.T.); koseimatsue@gmail.com (K.M.)
3    Kameda Medical Center, Department of Radiology, Chiba 296-0041, Japan; machida.yoichi@kameda.jp
*    Correspondence: kota1986ky@yahoo.co.jp; Tel.: +81-3-5803-5311; Fax: +81-3-5803-0147

**Abstract:** To clarify the prognostic value of 2-[$^{18}$F]fluoro-2-deoxy-D-glucose positron emission tomography/computed tomography ($^{18}$F-FDG PET/CT) in Neurolymphomatosis (NL), we retrospectively reviewed medical records of all NL patients who had undergone $^{18}$F-FDG PET/CT from 2007 to 2020 at Kameda Medical Center and Tokyo Medical and Dental University Hospital. The clinical data of patients were compared with $^{18}$F-FDG PET/CT findings of number of nerve lesions and presence of non-nerve extranodal lesions (ENL). Subsequently, we calculated recurrence-free survival (RFS) and overall survival (OS) using the Kaplan–Meier method. A total of 28 patients (mean age 70.1 years, range 44–87 years; 15 women) were included in the study and 7 patients (25.0%) relapsed NL. The number of nerve lesions detected by $^{18}$F-FDG PET/CT ranged from 1 to 5, with an average of 2.02. ENL was observed in 18 cases (64.3%). The comparison between the findings revealed that the more the lesions detected by $^{18}$F-FDG PET/CT, the higher the probability of recurrence ($\chi^2$ = 13.651, $p$ = 0.0085) and there was significantly shorter RFS for the patients with 3 or more nerve lesions ($p$ = 0.0059), whereas the presence of ENL was not significantly associated with any clinical findings. The present study revealed that the more nerve lesions detected by $^{18}$F-FDG PET/CT, the poorer the recurrence rate and RFS.

**Keywords:** neurolymphomatosis; malignant lymphoma; FDG PET/CT; 2-[$^{18}$F]fluoro-2-deoxy-D-glucose; prognostic value; recurrence; recurrence-free survival

## 1. Introduction

Neurolymphomatosis (NL) is a rare condition that occurs in patients with malignant lymphoma (ML) and is characterized by the direct invasion of the peripheral nervous system by neoplasm cells. It is most observed in patients with non-Hodgkin's B-cell lymphoma (NHL), especially in cases of diffuse large B-cell lymphoma (DLBCL) [1–3]. Although the exact prevalence of NL is unknown, it is estimated to occur in approximately 0.2% of all NHL patients [2]. The peripheral nerves are the main target of NL (60%), followed by the spinal nerves (48%), cranial nerves (46%), and plexus (40%) [1]. Common presentations include painful peripheral neuropathy or radiculopathy, cranial neuropathy, painless polyneuropathy, peripheral mononeuropathy, or mononeuropathy multiplex [1,4].

A definitive diagnosis of NL is based on pathological confirmation by nerve biopsy. However, tests that are highly invasive that risk permanent nerve damage are difficult to perform because, like other MLs, the entity can result in remission without sequelae if chemotherapy is successful. In addition, performing a biopsy is often difficult because NL frequently shows proximal and localized involvement [5,6]. Cerebrospinal fluid cytology also plays an auxiliary role in the diagnosis of central nerve infiltration, although it is less sensitive and inadequate for detecting peripheral nerve lesions [1,7,8]. Magnetic resonance

imaging (MRI) is a useful diagnostic tool for evaluating peripheral nerves in patients with NL and has been reported to have diagnostic accuracies as high as 77% [1]; however, some studies have reported accuracies as low as 40–59% [7,9], which is insufficient for a definitive diagnosis.

For assessing ML, the diagnostic and prognostic value of 2-[$^{18}$F]fluoro-2-deoxy-D-glucose positron emission tomography/computed tomography ($^{18}$F-FDG PET/CT) is well established. It has high sensitivity for detecting NL and can show nodular or linear uptake along the neural pathways. $^{18}$F-FDG PET/CT has remarkably high diagnostic accuracy, so a nerve biopsy is not always required if a pathological diagnosis of lymphoma is confirmed in other sites [1,7,9].

Because of the difficulties in diagnosing NL and the poor prognoses due to the clinical course of the underlying lymphoma, few studies have focused on the correlation between imaging findings and clinical outcomes of NL. Particularly, to the best of our knowledge, there are no reports focused on the imaging findings of NL with recurrence. Therefore, we retrospectively reviewed the medical records and $^{18}$F-FDG PET/CT findings of patients with NL to clarify the correlation between imaging and clinical outcomes, with a focus on the likelihood of the recurrence. In this study, recurrence is specifically defined as NL recurrence, not ML recurrence.

## 1.1. Subjects and Data Extraction

We retrospectively reviewed the medical records and imaging findings of all NL patients who had undergone $^{18}$F-FDG PET/CT between February 2007 and December 2020 in Kameda Medical Center (KMC) and Tokyo Medical and Dental University Hospital (TMDUH). Diagnosis of NL was based on clinical and/or pathological findings from imaging findings, which included $^{18}$F-FDG PET/CT. We collected data on age, sex, underlying lymphoma, and post-treatment course from patient medical records. We also searched for recurrence, death date, or last visit date to calculate recurrence-free survival (RFS) and overall survival (OS).

Two nuclear medicine experts assessed all serial $^{18}$F-FDG PET/CT images for number and location of affected nerves and absence/presence of non-nerve extranodal lesions (ENL). If a patient had undergone $^{18}$F-FDG PET/CT more than once, we used the image that was acquired at the time of NL diagnosis for the analysis. MRI was used as a reference to confirm that sites with suspected $^{18}$F-FDG accumulations, such as cranial nerve lesions, were true lesions.

## 1.2. $^{18}$F-FDG PET/CT Protocol

All patients fasted for at least 6 h prior to undergoing PET/CT imaging. In all patients, we confirmed a blood glucose level under 180 mg/dl before intravenously injecting with 3.7 MBq/kg of $^{18}$F-FDG. One hour after $^{18}$F-FDG injection, patients underwent PET/CT imaging. Image acquisition was performed from the top of the head to the middle of the thigh. Attenuation-corrected PET, non-attenuation-corrected PET, and CT images were reviewed, and the attenuation-corrected PET images were coregistered to the CT images.

## 1.3. Statistical Analysis

The software packages SPSS (version 27; IBM Corp. Armonk, NY, USA) and MedCalc (version 11.6; MedCalc Software) were used for statistical analyses. Continuous variables were compared using a t test and categorical variables were compared using a chi-square test or Fisher's exact probability method to investigate the correlation between the clinical and imaging findings. Receiver operating characteristic (ROC) analysis was performed to predict prognosis by continuous variables. The Kaplan–Meier method was used for OS and RFS analyses. Multiple Cox regression analysis was used to analyze the factors of recurrence and survival. The log-rank test was used to compare OS and RFS between different groups classified based on $^{18}$F-FDG PET/CT findings. Cox proportional hazards regression analyses were performed to assess the prognostic value of the different classifiers

(categorical predictors). Results were considered statistically significant if $p < 0.05$. OS was defined as the time interval between the $^{18}$F-FDG PET/CT scan at the time of NL diagnosis and death by any cause, whereas RFS was defined as the time interval between the PET/CT scan and the first recurrence of NL or death by any cause.

## 2. Results

### 2.1. Patient Characteristics and $^{18}$F-FDG PET/CT Findings

Of the 28 patients included in the study, 24 were from KMC and the remaining four were from TMDUH. Patient characteristics are shown in Table 1. The mean age was 70.1 years (range 44–87 years) and 15 were women (53.6%). Patients had the following underlining lymphomas: 21 patients had DLBCL (75%), 5 had intravascular lymphomas (IVL) (17.9%), and 2 had mantle cell lymphomas (MCL) (7.1%). Of all 28 patients, 7 patients (25.0%) had a relapse of NL and 14 patients (50.0%) died within the observational period. Of the 7 cases who had a recurrence, 5 had DLBCL, 1 had IVL, and 1 had MCL. The number of the nerve lesions observed in the $^{18}$F-FDG PET/CT assessment ranged from 1 to 5, with an average of 2.04 lesions. The trigeminal nerves and cauda equina to the lumbar plexus were the most common sites detected by $^{18}$F-FDG PET/CT (Figure 1). Non-nerve ENLs were observed in 18 cases (64.3%).

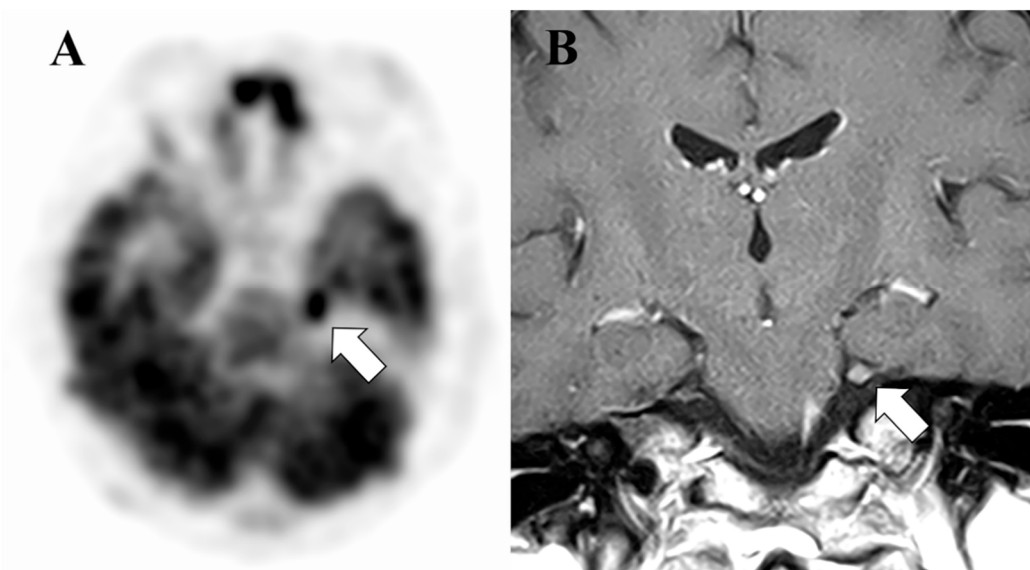

**Figure 1.** A 43-year-old female patient with NL. (**A**). A FDG PET image shows high accumulation in the left trigeminal nerve. (**B**). A coronal image of contrast enhanced MRI shows thickening and abnormal enhancement of the left trigeminal nerve, which was confirmed to be a true lesion.

**Table 1.** Clinical characteristics of all 28 patients.

| Clinical Findings | |
| --- | --- |
| Age, years; mean $\pm$ SD (range) | 70.1 $\pm$ 10.1 (44–87) |
| Sex, n (%) | |
| Female | 15 (53.6) |
| Male | 13 (46.4) |
| Subtype of lymphoma, n (%) | |
| DLBCL | 21 (75.0) |
| IVL | 5 (17.9) |
| MCL | 2 (7.1) |
| IPI at the initial diagnosis of lymphoma | |
| 5 | 3 (10.7) |
| 4 | 15 (53.6) |
| 3 | 5 (17.9) |
| 2 | 2 (7.1) |
| 1 | 1 (3.6) |
| 0 | 0 (0) |
| NA | 2 (7.1) |
| FDG PET/CT findings, n (%) | |
| Number of nerve lesions | |
| <3 | 19 (67.9) |
| ⩾3 | 9 (32.1) |
| ENL | |
| + | 18 (64.3) |
| − | 10 (35.7) |
| Outcome, n (%) | |
| Recurrence | |
| + | 7 (25.0) |
| − | 21 (75.0) |
| Status at final follow-up | |
| Dead | 14 (50.0) |
| Alive | 14 (50.0) |

n—number of patients, ENL—extranodal lesions, DLBCL—diffuse large B-cell lymphoma, IVL—intravascular lymphoma, MCL—mantle cell lymphoma, IPI—International Prognostic Index., NA—not available.

## 2.2. Comparison between Clinical and $^{18}F$-FDG PET/CT Findings

Results of the comparison between the clinical and $^{18}$F-FDG PET/CT findings are summarized in Table 2. The most notable result was that in patients who relapsed during the course, we detected more neural lesions on their initial $^{18}$F-FDG PET/CT. Recurrence was more likely to occur when the number of nerve lesions detected by $^{18}$F-FDG PET/CT was 3 or more ($\chi^2$ = 11.842, $p$ = 0.0006), and ROC analysis showed that $^{18}$F-FDG PET/CT findings were useful in predicting prognosis with a sensitivity of 85.7, specificity of 85.7, and AUC = 0.91 ($p$ < 0.001) when the number of nerve lesions was used to predict NL recurrence (Figure 2). $^{18}$F-FDG PET/CT findings were shown to be useful in predicting prognosis. A scatter plot of the recurrence rate according to the number of affected nerves and a hypothetical regression line show that the higher the number of nerve lesions, the

more likely the recurrence (Figure 3). However, due to the small number of cases, the results are not statistically significant. [18]F-FDG PET/CT images of cases with recurrence are shown in Figure 4. There was no statistically significant association between the presence or absence of ENL and clinical findings. All IVL and MCL cases had ENL, whereas 10 of 21 DLBCL cases did not have ENL, though there were no statistically significant differences ($p = 0.0748$).

**Table 2.** Comparison of clinical and FDG PET/CT findings.

| | FDG PET/CT Findings | | | | | |
|---|---|---|---|---|---|---|
| | Number of Nerve Lesions | | | ENL | | |
| | <3 (*n* = 19) | ⩾3 (*n* = 9) | *p* | + (*n* = 18) | − (*n* = 10) | *p* |
| Age, years; mean ± SD | 72.9 ± 11.1 | 70.4 ± 8.8 | 0.5593 | 74.3 ± 8.1 | 68.3 ± 13.1 | 0.1446 |
| Sex, n (%) | | | 0.8869 | | | 0.7815 |
| Female | 10 (52.6) | 5 (55.6) | | 10 (55.6) | 5 (50) | |
| Male | 9 (47.4) | 4 (44.4) | | 8 (44.4) | 5 (50) | |
| Subtype of ML, n (%) | | | 0.757 | | | 0.0748 |
| DLBCL | 15 (71.4) | 6 (28.6) | | 11 (52.3) | 10 (47.7) | |
| IVL | 3 (60) | 2 (40) | | 5 (100) | 0 (0.0) | |
| MCL | 1 (50) | 1 (50) | | 2 (100) | 0 (0.0) | |
| Recurrence, n (%) | | | 0.0085 | | | 0.6547 |
| + | 1 (14.3) | 6 (85.7) | | 4 (57.1) | 3 (42.9) | |
| − | 18 (85.7) | 3 (14.3) | | 14 (66.7) | 7 (33.3) | |

ENL—extranodal lesions, ML—malignant lymphoma, DLBCL—diffuse large B-cell lymphoma, IVL—intravascular lymphoma, MCL—mantle cell lymphoma.

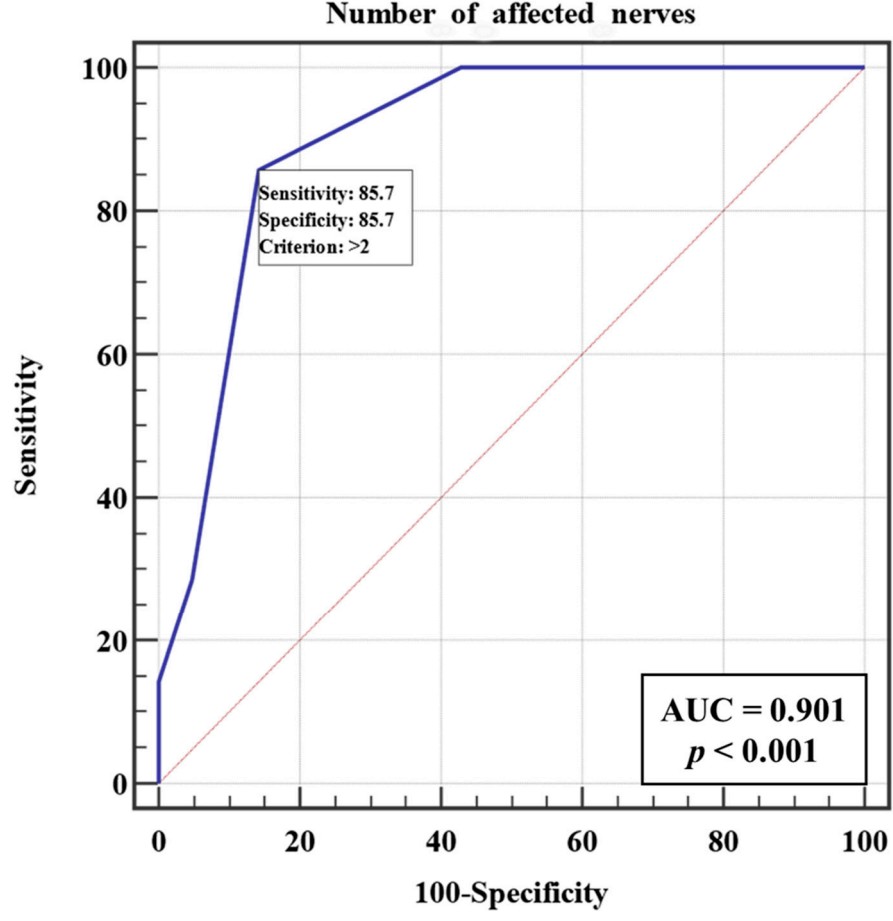

**Figure 2.** ROC curve for prediction of NL recurrence by number of affected nerves.

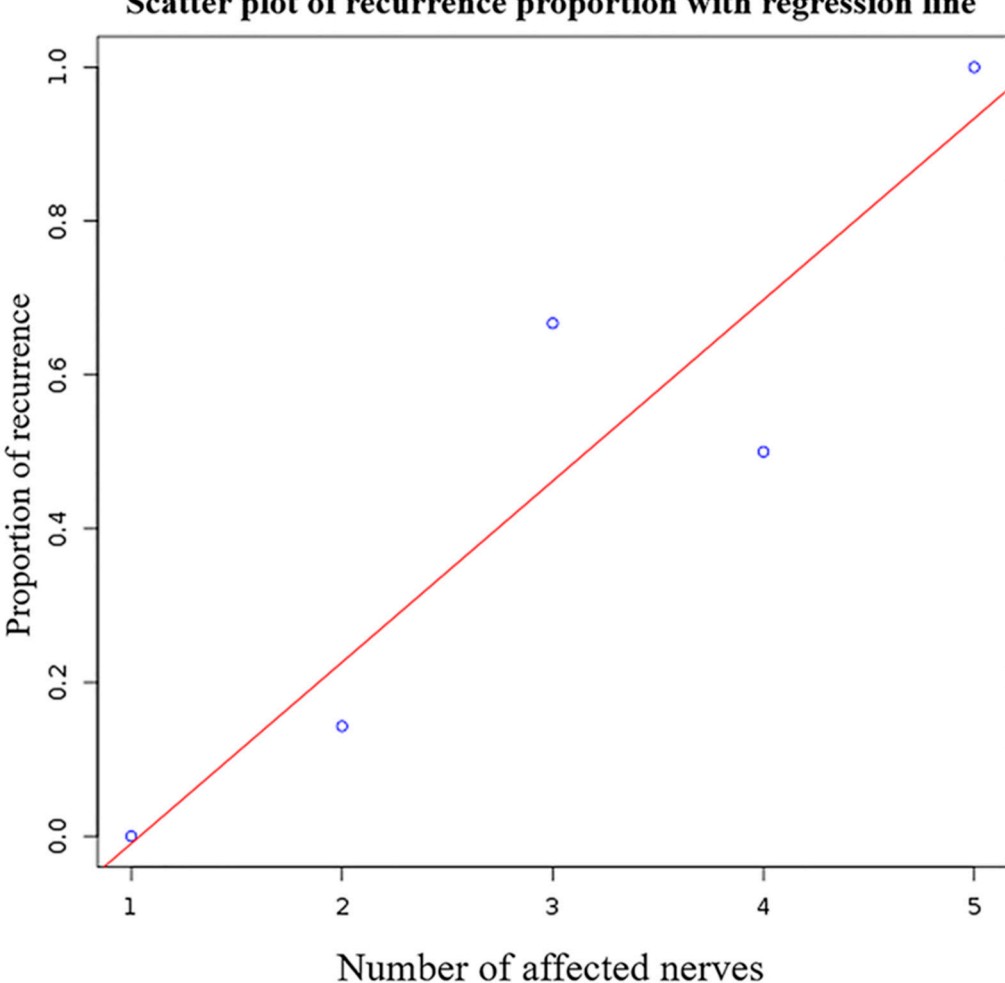

**Figure 3.** Scatter plot of recurrence proportion by number of affected nerves with regression line.

The graph shows that the more nerves affected, the more likely it is to recur, although the results are not statistically significant, due to the small number of cases.

*2.3. RFS and OS Analyses*

A total of 14 patients (50%) died during follow up and 14 patients were still alive at final contact. The median follow-up period for NL was 14.6 mo (range 0.6–166.7 mo) and the mean follow-up period was 24.53 mo (±standard deviation [SD] 33.12 mo). The median RFS period was 12.7 mo (range 0.6 to 166.7 mo) and the mean RFS period was 22.0 mo (±SD 33.41 mo).

The diagnostic groups classified based on the [18]F-FDG PET/CT findings for the Kaplan–Meier analyses and log-rank test were as follows: patients with fewer than three or three or more neural lesions and patients with absence or presence of ENL. The cut-off value for the number of neural lesions was determined from the ROC analysis (Figure 2). The Kaplan–Meier analysis and log-rank test revealed that patients with three or more neural lesions detected by [18]F-FDG/CT at the time of NL diagnosis had significantly shorter RFS ($p = 0.0059$, hazard ratio [HR] 4.985, 95% confidence interval [CI] 1.590–15.634; Figure 5), whereas OS did not significantly differ between groups ($p = 0.063$, HR 2.87, 95% CI 0.944–8.741; Figure 6). The presence or absence of ENL did not significantly differ for OS or RFS.

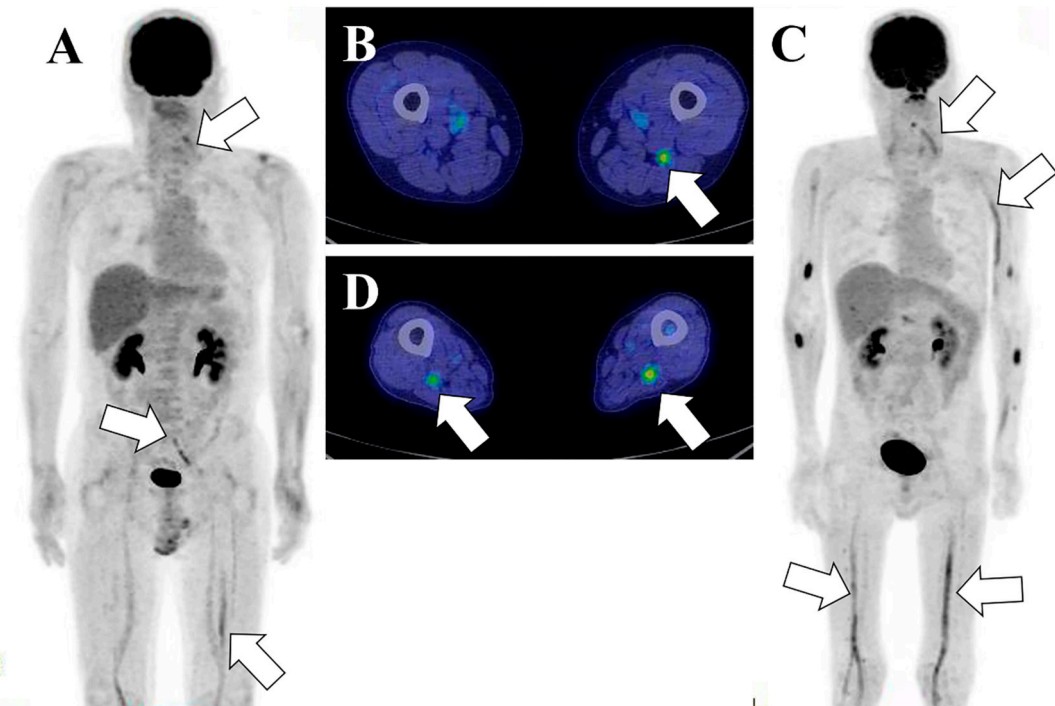

**Figure 4.** A 60-year-old male patient with NL who relapsed 2 months later. (**A**,**B**). Initial FDG PET/CT image at time of diagnosis. Maximum intensity projection (MIP) image of the FDG PET image demonstrating hypermetabolic lesions in the lumbar plexus, left sciatic nerve, and left cervical spinal nerve 6 (C6) (**A**). PET/CT fusion image shows a hypermetabolic lesion in the left sciatic nerve (**B**). (**C**,**D**). FDG PET/CT images two months later at the time of NL recurrence. The MIP image shows intense hypermetabolic lesions along bilateral sciatic nerves, left C6 nerve to brachial plexus, and left trigeminal nerve, while the lumber plexus lesion has disappeared. Nodular lesions around the elbow joints are extranodal lesions outside the neural structure (**C**). PET/CT fusion image shows hypermetabolic lesions in bilateral sciatic nerves (**D**).

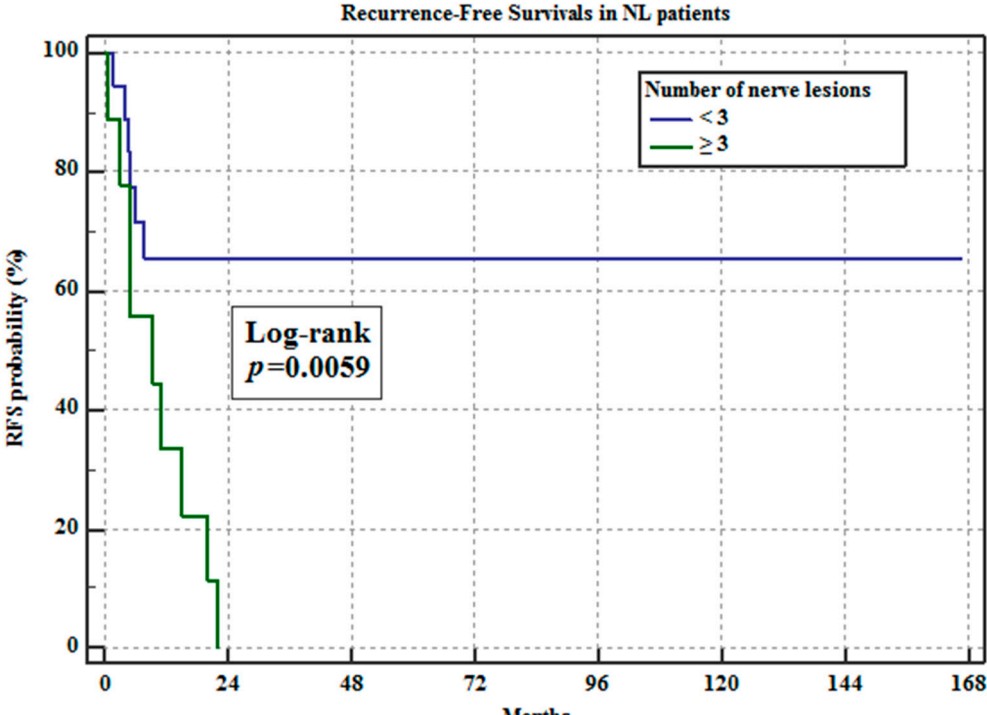

**Figure 5.** Kaplan–Meier analysis for RFS.

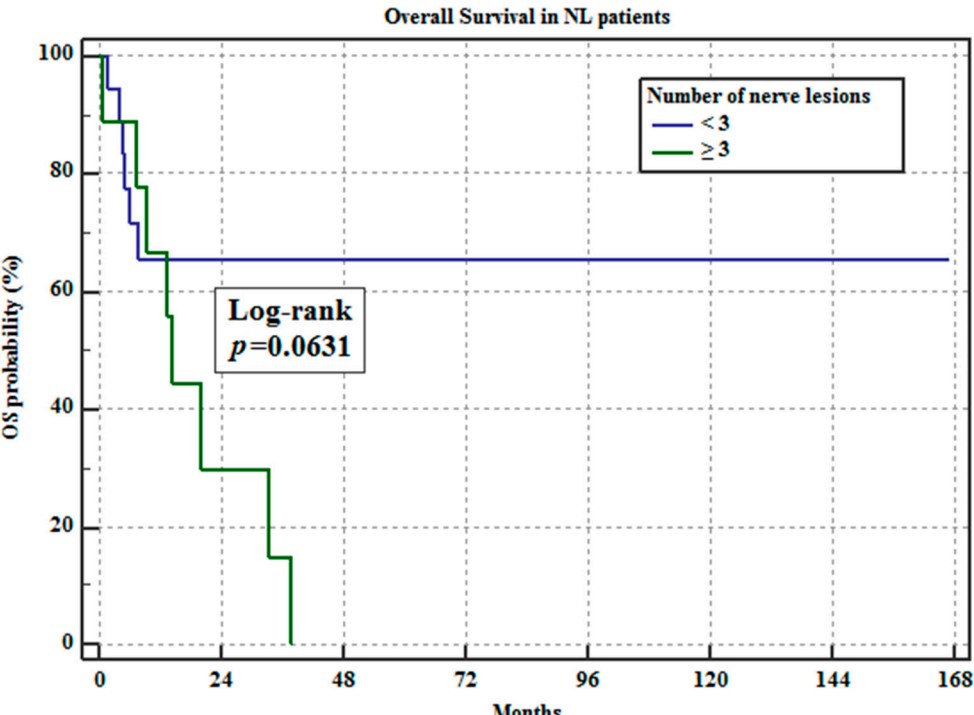

**Figure 6.** Kaplan–Meier analysis for OS.

Patients with three or more neural lesions detected by [18]F-FDG/CT had significantly shorter RFS (*p* = 0.0059 for log-rank test).

Patients with three or more neural lesions detected by [18]F-FDG/CT had shorter OS; however, the difference was not significant (*p* = 0.0631 for log-rank test).

## 3. Discussion

The present study revealed that in patients with NL, the more nerve lesions that were detected by [18]F-FDG PET/CT, the more likely that NL recurred. Moreover, RFS tended to be shorter in patients with three or more nerve lesions detected by [18]F-FDG PET/CT. NL is considered a rare condition and occurs in patients with progressive ML. It has a poor prognosis [1,10,11], with a previous study reporting one- and three-year survival rates of 46% and 24%, respectively [1]. Furthermore, because the nerve biopsy required for a definitive diagnosis of NL is highly invasive, many patients die due to the underlying ML without being diagnosed with NL, and there are few reports on the clinical course of NL. However, some studies have noted that the increased recognition of the utility of [18]F-FDG PET/CT for NL diagnosis may suggest higher prevalence than estimated [2,7,12]. Because [18]F-FDG PET/CT has high sensitivity for detecting active lesions in NL, biopsies are not necessary. Our study revealed that the clinical course of NL can be determined using [18]F-FDG PET/CT findings. Above all, the most valuable element of our study is that we were able to compare the recurrence cases, which are rarely reported, with the non-recurrence cases. In regard to the background lymphoma in patients with NL, DLBCL was the most common in our study at 75%, which is consistent with a previous and largest case series to date that reported DLBCL in 73.5% of 50 cases [1]. Other histological subtypes observed in the present study were IVL and MCL. Previous studies have reported follicular lymphoma, peripheral T-cell lymphoma, MCL [11], IVL [13], Burkitt's lymphoma [14], and NK-cell lymphoma [15] as underlying diseases. We did not find any significant differences between the subtypes of underlying lymphoma. However, we did not have a sufficient sample of IVL or MCL patients to make statistical comparisons. These lymphomas are rarer than DLBCL and require further study in a larger number of cases. It has been reported that 58% of NL patients have multiple nerve lesions; in our study, we found multiple lesions

in 57.1% of patients. There have been no reports on the association between presence of multiple lesions and prognosis or clinical course and we newly found that the number of affected nerves detected by $^{18}$F-FDG PET/CT in NL patients significantly correlated with the likelihood of recurrence. The presence of ENL is one of the factors leading to poor prognosis of ML; however, we found that it did not significantly correlate with the clinical course of NL. The median OS from the time of diagnosis of NL was 14.6 months in our study, which is consistent with previous reports of 10–15 months [1,7,11].

This study has several limitations. Firstly, NL is a phenotype of uncontrolled ML that shows poor prognosis, and many patients may die due to the underlying lymphoma. Therefore, OS is greatly affected by the prognosis of the underlying lymphoma or other complications. We estimated RFS from the time of diagnosis, but the recurrence-free group included cases of poor prognosis who died of the primary disease within a short period without achieving remission. It is clear that the control of the primary disease is a major confounding factor, which may explain why presence of ENL did not correlate with prognosis. There are confounding factors that affect the prognosis of malignant lymphoma, such as treatment methods.

In this study, standard treatment according to guidelines was used. It would be desirable to confirm the correlation with prognosis and recurrence rate by multivariate analysis, but we did not have a sufficient number of cases to perform multivariate analysis. There have been few reports investigating the recurrence of NL, and this study was useful in that regard. However, we only had five cases of recurrence, thus further studies with more cases are necessary.

The lack of histological verification of NL diagnosis is another limitation given that there have been previous reports of lymphoma patients with other clinical entities that involve peripheral nerves [16,17]. The various underlying mechanisms include neurotoxicity secondary to chemotherapy; infections; immune-mediated, paraneoplastic or metabolic processes; and nutritional deficiencies [17]. However, several studies have shown that $^{18}$F-FDG PET/CT findings, combined with clinical manifestation information, have the highest yield for antemortem NL diagnosis; moreover, biopsy is not always necessary [1,9,10,18]. For PET/CT analysis, we used qualitative indices, such as the number of nerve lesions and presence or absence of ENL, but a comparison with quantitative indices using standardized uptake value is also warranted. MRI is another useful imaging tool for evaluating peripheral nerves. NL findings include diffuse or nodular thickening of nerves, T2 prolongation, and abnormal enhancement of nerve lesions [3,8]. We did not include these findings because recent studies of NL have reported diagnostic accuracies of 59–77% for MRI, which is inferior to those of $^{18}$F-FDG PET/CT, which have been reported to be 84–100% [1,9]. In particular, the detectability of small lesions is said to be lower than that of $^{18}$F-FDG PET/CT [18,19]. However, there is value in using $^{18}$F-FDG PET/CT and MRI in combination in clinical settings and comparing the combined data with clinical information. Furthermore, flow cytometry, immunoglobulin heavy chain rearrangement, and microRNA are other new methods that have the potential to be useful in the diagnosis of CNS lesions, but their usefulness in the detection of peripheral nerve lesions has not yet been reported [1,7,8,20,21]. In the future, it will be important to compare these and $^{18}$F-FDG PET/CT and/or MRI findings to detect biomarkers related to the diagnosis and prognosis of NL. Finally, strong prognostic factors of ML, such as the international prognostic index, should be considered, but in this study, $^{18}$F-FDG PET/CT was used as a criterion for both the diagnosis and recurrence of NL, and the recurrence of background ML was not considered. Further studies are needed to establish evidence for the prognostic value of IPI and LDH combined with $^{18}$F-FDG PET/CT findings, especially for OS. Although the evidence is still limited, we believe that $^{18}$F-FDG PET/CT is promising for the prognostic evaluation of NL.

## 4. Conclusions

We revealed that the more neural lesions detected by $^{18}$F-FDG PET/CT at the time of NL diagnosis, the higher the recurrence rate and the worse the RFS.

**Author Contributions:** Conceptualization, K.M. and U.T.; methodology, K.Y. and U.T.; formal analysis, K.Y., U.T.; investigation, K.Y., J.T., T.T., Y.M., K.M. and U.T.; data curation, M.U., K.Y., Y.M. and U.T.; writing—original draft preparation, M.U. and K.Y.; writing—review and editing, J.T., T.T., Y.M., K.M. and U.T.; supervision, U.T.; funding acquisition, U.T. All authors have read and agreed to the published version of the manuscript.

**Funding:** This study was supported in part by the National Cancer Center Research and Development Fund (2020-J-3). This study was also supported by the Ministry of Health, Labour and Welfare Grants and Japan Agency for Medical Research and Development Grants No.15ck0106162h0001, 16ck0106162h0002, 20ck0106577h0001, and 17ck0106162h0003.

**Institutional Review Board Statement:** The study was conducted according to the guidelines of the Declaration of Helsinki, and approved by the Institutional Review Board (or Ethics Committee) of Tokyo Medical and Dental University ((M2021-095 (19 July 2021)).

**Informed Consent Statement:** Written informed consent has been obtained from the patients to publish this paper.

**Data Availability Statement:** The data presented in this study are available on request from the corresponding author. The data are not publicly available due to the ethics committee has not approved the release of detailed data.

**Acknowledgments:** The authors thank Guideline Committee of Japan Radiological Society (JRS) for their vulnerable assistance in acquisition of article and for their helpful suggestions. We have also been given helpful suggestions by Daniel C Sullivan, and Alexander Guimaraes, Chair and Steering Committee, Quantitative Imaging Biomarkers Alliance (QIBA), Radiological Society of North America (RSNA) and Shigeki Aoki, and Ukihide Tateishi, Chair and Steering Committee, J-QIBA, JRS.

**Conflicts of Interest:** The authors declare no conflict of interest.

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
