# Peer review of "Impact of 18F-FDG PET/CT in Predicting Recurrence in Neurolymphomatosis"

_hemato, doi:10.3390/hemato2030038_

Round 1

Reviewer 1 Report

please find the enclosed attatched file

Author Response

General Comments

We thank the reviewer for the helpful comments and suggestions, which we believe have strengthened the manuscript substantially. We have addressed all comments in our revised manuscript and discuss them in the attached file on a point-by-point basis. All changes to the revised manuscript are presented in red font in the manuscript.

Reviewer 2 Report

The authors reported a series of neurolymphomatosis (28 patients). Most patients had diffuse large B cell lymphoma. According to their analysis, the number of nerve lesions adversely affect patient outcome in terms of recurrence free survival and possible overall survival by trend.

Such data is precious and worth clinical attention.

I have several comments and questions

  1. In table 2, the patient numbers add up but the percentage doesn’t. The gender, lymphoma subtypes and recurrence percentages are all confusing. It is a serious problem which makes the statistical results questionable.
  2. Prognosis of lymphoma is determined by multiple factors. The number of ENL is a prognostic factor identified by authors. It needs to be correlated to other factors, such as LDH, stage, number of total extranodal lesions. The international prognostic index (IPI) is the most widely used model in predicting prognosis. The authors should include such scores and clinical parameters in analysis and correlation.
  3. The cut-off of nerve lesion number is set as 3. While statistical significance is important, the author should give reasons of such a cutoff point.
  4. The number in table 2 and figure 5 do not match. In table 2, only one patient had recurrence among 19 patients with ENL number <3. In figure 5, more than 30% of patient had recurrence and in fact, succumbed to the disease according to the OS curve.
  5. The prognosis is strongly affected by the treatment they received. The authors should at least mention the kind of treatment they received. In particular, most patients are elderly. Did they receive R-CHOP treatment? In view of their neurological involvement, did they receive intrathecal injection or high dose methotrexate/cytarabine ?
  6. I think they should report the sites of recurrence. Do they always recur in the nervous system?

Author Response

(The authors gave the same response as above.)

Reviewer 3 Report

This paper from Megumi Uemichi et al provides interesting data on the PET-guided evaluation of neurolymphomatosis. As the authors indicate, the correlation of PET avid lesions in central nervous system and clinical outcome has not been performed previously. Therefore, although this work is limited by the low number of patients, observations provided here are valuable and of interest. My major concern is about statistics usage. Because of the limited number of events, Chi square may not be suitable, and a Fisher exact test should be performed instead. In the same line, the authors attempt to show a linear relationship between the number of afected nerves and outcome, however there are only seven cases of progression, so the model would be rather unreliable. The authors should either discuss this more deeply or modify this results section. Related to the previous point, association is not prediction (which would be a desirable goal), so maybe ROC analysis should be performed for the variables associated with progression and c-statistics should be reported. Finally, the authors recognise some of the limitations of the paper, which are mainly comparing the technique to other imaging techiniques. However, other groups have proposed other biological studies by lumbar puncture such as flow cytometry, VDJ rearrangements or miRNA for the diagnosis of leptomeningeal infiltration or even deep CNS infiltration. The authors should provide further information on this.

Author Response

General Comments

We thank the reviewer for the helpful comments and suggestions, which we believe have strengthened the manuscript substantially. We have addressed all comments in our revised manuscript and discuss them below on a point-by-point basis. All changes to the revised manuscript are presented in red font in the manuscript.

Round 2

Reviewer 2 Report

The authors provided quite some explanations and several minor revisions after my review opinions. While I find some drawbacks, I would like to emphasize that this data is previous and important. Such a material is well worth publication. Still, I find some issues and concerns.

1#I am surprised to find the authors defined recurrence as recurrence of NL rather than lymphoma. According to general understanding and what is described in the manuscript. NL is characterized by the direct invasion of the peripheral nervous system by neoplasm cells. That means recurrence of NL is definitive recurrence of lymphoma although not vice versa. I am not sure how we should look at this disparity. If the authors insist on their own definition, they should at least describe their definition of recurrence (of NL) in the article.

2#As for the risk factors of lymphoma relapse (relapse is used more commonly than recurrence for lymphoma), such as IPI, LDH, stage, they are usually defined at initial diagnosis of lymphoma. The authors don’t have to collect data during subsequent timing during the clinical course. In many centers, such fundamental workup, at least staging of cancer before beginning treatment is a standard practice.

3#There are still problems in table 2. The DLBCL percentage doesn’t add up. In addition, I would like to raise a logic issue of scientific presentations. The authors calculated percentage horizontally. For example, in gender analysis, this means in female patients 66.7% had NL lesion number<3 and 33.3% had NL lesion number >=3. It doesn’t really make sense. In such an analysis, you want to see if the baseline characteristics are different between two groups (lesion number<3 and ≧3). It should be presented as female 52.6% and male 47.4% for lesion number<3.
